# MKD: Mixup-Based Knowledge Distillation for Mandarin End-to-End Speech Recognition

**Xing Wu** [1,2,3,*] , **Yifan Jin** [1] , **Jianjia Wang** [1,2] , **Quan Qian** [1,2,3] **and Yike Guo** [4]

1   School of Computer Engineering and Science, Shanghai University, Shanghai 200444, China;
    jjyyff@shu.edu.cn (Y.J.); jianjiawang@shu.edu.cn (J.W.); qqian@staff.shu.edu.cn (Q.Q.)
2   Shanghai Institute for Advanced Communication and Data Science, Shanghai University,
    Shanghai 200444, China
3   Materials Genome Institute, Shanghai University, Shanghai 200444, China
4   Department of Computer Science, Hong Kong Baptist University, Hong Kong 999077, China;
    yikeguo@hkbu.edu.hk
*   Correspondence: xingwu@shu.edu.cn

**Abstract:** Large-scale automatic speech recognition model has achieved impressive performance. However, huge computational resources and massive amount of data are required to train an ASR model. Knowledge distillation is a prevalent model compression method which transfers the knowledge from large model to small model. To improve the efficiency of knowledge distillation for end-to-end speech recognition especially in the low-resource setting, a Mixup-based Knowledge Distillation (MKD) method is proposed which combines Mixup, a data-agnostic data augmentation method, with softmax-level knowledge distillation. A loss-level mixture is presented to address the problem caused by the non-linearity of label in the KL-divergence when adopting Mixup to the teacher–student framework. It is mathematically shown that optimizing the mixture of loss function is equivalent to optimize an upper bound of the original knowledge distillation loss. The proposed MKD takes the advantage of Mixup and brings robustness to the model even with a small amount of training data. The experiments on Aishell-1 show that MKD obtains a 15.6% and 3.3% relative improvement on two student models with different parameter scales compared with the existing methods. Experiments on data efficiency demonstrate MKD achieves similar results with only half of the original dataset.

**Keywords:** end-to-end speech recognition; knowledge distillation; model compression; data efficiency; mixup

## 1. Introduction

Deep neural networks have been successfully applied to the field of speech recognition. In recent years, Transformer-based speech recognition models [1] have gradually become mainstream. The performance of speech recognition model based on Transformer has been greatly improved compared with previous CNNs [2] and RNNs [3], but its computational complexity has increased significantly either. A high-precision speech recognition model usually has a parameter scale of tens of millions or even billions, which requires huge computational resources and storage space. For some devices with low computing power, such as: mobile devices, edge computing devices, etc., it is impossible to deploy large models. Therefore, model compression is needed to reduce the arithmetic requirements of the models for the deployed devices.

Knowledge Distillation (KD) [4] is a popular model compression method which transfers the knowledge of teacher model to student model. The purpose of KD is making the student model mimic the behavior of the teacher model through soft labels. The soft labels not only contain correct category distribution, but also reflect the relationship between similar categories which improves the efficiency of training. Previous studies [5,6] have

focused on the mode of distillation. However, much fewer studies have optimized the distillation efficiency from the perspective of data. In fact, when the size of training dataset is small, the generalization of student models obtained by distillation will not be strong enough [7]. The distribution of dataset has inability to fit the true distribution correctly, which can easily lead to overfitting problems.

Data augmentation is a method to prevent model overfitting by adding perturbations to the original data, which encourages the model to learn more robust features. With the introduction of audio data augmentation methods such as Specaugment [8] and Mixspeech [9], audio data augmentation has become the simplest and the most efficient method to improve speech recognition accuracy. Among these data augmentation methods, Mixup [10] is the most suitable for few-shot learning because of its ability to simulate real distribution. The advantage of data agnosticism becomes another reason to choose Mixup as the candidate.

To improve the data efficiency of knowledge distillation, a Mixup-based Knowledge Distillation framework named MKD is proposed for end-to-end speech recognition by combining Mixup with knowledge distillation. Frame-level fusion of soft label is employed when applying Mixup to knowledge distillation. For speech recognition, the weighted fusion of soft label sequence cannot be performed directly because the label sequence is discrete. The fusion at the loss function level becomes an alternative choice. Further, a new loss function is proposed to apply Mixup to Kullback–Leibler (KL) divergence. It is proved mathematically that optimizing the new loss function is equivalent to optimizing the upper bound of the original loss function. It is experimentally demonstrated that on Aishell-1 [11]. Two student models with different parameter scale are trained with the proposed MKD. The model with half the size of teacher model achieves a 15.6% improvement, and another student model with fewer parameters achieves a 3.3% improvement compared to the softmax-level knowledge distillation method.

To best of our knowledge, our approach is the first attempt to combine Mixup approach with knowledge distillation for ASR. The contributions of this paper can be summarized in the following three points.

- A knowledge distillation framework named MKD is proposed by combining Mixup with softmax-level knowledge distillation for end-to-end speech recognition.
- A mixed loss function $\mathcal{L}_{MKD}$ is proposed based on KL-divergence, and it is theoretically shown that optimizing $\mathcal{L}_{MKD}$ is equivalent to optimize an upper bound of the original knowledge distillation loss.
- Experimentally, our proposed MKD beats the original method on Aishell-1. The model with half the size of teacher model achieves a 15.6% improvement, and another student model with fewer parameters achieves a 3.3% improvement. Experiments of data efficiency show the advantages of MKD under a limited-data setting.

## 2. Related Work

Deep neural networks have achieved great success in many fields such as [12,13], especially in automatic speech recognition [14]. End-to-end speech recognition has become the mainstream method for training ASR models. The size of end-to-end speech recognition models is huge not matter which framework is adopted from CNN [15], RNN [16] or Transformer [17]. However, it requires heavy computation for both training and testing when the model architecture gets deeper. To mitigate this computational burden, there has been a long list of research on model compression. Knowledge Distillation [18] has been demonstrated as an efficient model compression method, which aims at transferring knowledge from a well-trained teacher model to a small student model. With this additional transfer procedure, the student model can perform better compared to naive training. Transferring class probability and transferring the representation of the hidden layer are two typical training strategies proposed from previous research.

Hinton [4] first introduced the concept of knowledge distillation by minimising the KL-divergence of the softmax outputs of teacher and student model. Generally, the output

layer of a classification model adopts softmax as the activation function. The output of softmax is a probability distribution over the label classes, and the sum of the outputs equals 1. The softmax prediction of teacher model has a nonzero probability value for each target class. This soft label is normally considered more informative than the one-hot encoded ground truth. The KD technique mentioned above only considers the output of the teacher model.

In the case of transferring the hidden representation, some KD methods [19,20] proposed transferring the representation-level information of the hidden layers which minimizes the mean squared error between the representation vector. For speech recognition tasks, Li [21] applied the teacher–student framework to achieve speech knowledge distillation for the first time. Geras [22] investigated heterogeneous knowledge distillation by refining knowledge from RNN-based models into CNN-based models, and Kurata [23] optimised a CTC-based [24] heterogeneous knowledge distillation method. Takashima [25] proposed sequence-level knowledge distillation in the CTC framework using the N-best hypotheses of teacher model. Wong [26,27] migrated sequence-level knowledge into a DNN-HMM framework, and Kim [28] optimized sequence-level distillation based on the probability of the output sequences.

Data augmentation is a popular tool for training speech recognition models, especially for low-source ASR [29]. The purpose of data augmentation is to constrain the overfitting problem by constructing additional new samples. The traditional data augmentation methods for speech recognition modify the raw audio. Kanda [30] investigated three distortion methods: vocal tract length distortion, speech rate distortion, and frequency-axis random distortion. Ko [31] changed the speed of the audio signal, producing three versions of the original signal with speed factors of 0.9, 1.0 and 1.1. Jaitly [32] imposed a random linear warping along the frequency dimension. Data augmentation schemes [33] was explored for low resource languages. These methods obtain a high increment of accuracy and have low implementation costs for ASR models. SpecAugment untied the model structure and the masked strategy. It regards the masked method as a means of data augmentation and performs random masking on the log-Mel spectrogram of the input speech. A random permuted method SpecSwap was presented to construct new samples. These noise-based methods effectively improve the robustness of the model.

More recently, some researchers focused on automatic data augmentation. AutoAugment [34] was proposed to learn a constant policy under a meta-learning setting for many image recognition tasks. Adversarial AutoAugment [35] improved AutoAugment by searching a policy resulting in a higher training loss. Lim [36] improved the policy search time by learning an efficient search strategy depend on density matching. A simplified and lossless automatic policy search method was mentioned in [37]. Kim [38] proposed Local Augment, which highly alters the local bias property. Lin [39] introduced a set of common geometric operations into training and testing images to improve the efficiency of data augmentation. In speech recognition, Park [40] modified the SpecAugment to adapts the length of the utterance. Three on-the-fly data augmentation methods [41] were proposed for sequence-to-sequence speech recognition. A sample-adaptive policy that perturbs the training samples based on the current loss value of the sample was investigated in [42].

## 3. Methodology and Aim

This section introduces the Mixup-based knowledge distillation, with Speech-Transform-er as the baseline. Firstly, the Transformer-based ASR model is introduced. Then, the detail of how to integrate Mixup into ASR model is explained. Finally the Mixup-based knowledge distillation method is proposed.

### 3.1. Speech-Transformer

In this work, Speech-Transformer is chosen as the baseline for $Model_T$ and $Model_S$. Speech-Transformer is a Transformer-based neural network for speech recognition which consists of encoder $E$ and decoder $D$. The spectrogram $X$ serves as the input of $E$. Each $X$

is a two-dimensional matrix whose size is $F * T$, where $F$ is the number of frequency and $T$ denotes the number of frame. $E$ encodes the spectrogram by self-attention mechanism. The result of $E$ is a two-dimensional embedding $e \in R^{d*T}$. $D$ decodes the embedding sequence $e$ based on the corresponding label sequence $Y$. Finally, the decoder output goes through a Softmax classifier to generate the probability distribution of each character. The whole process can be summarized as:

$$
\begin{aligned}
E &: X \to f_1 \to f_2 \to \cdots \to f_n \to e \\
D &: (e, Y) \to g_1 \to g_2 \to \cdots \to g_m \to e_d \\
C &: e_d \to Softmax \to y
\end{aligned}
\tag{1}
$$

where $f_1, f_2, \cdots, f_n$ represent the hidden layers in encoder $E$. $g_1, g_2, \cdots, g_m$ are the hidden layers of decoder $D$. $y$ denotes the one-hot encoding of a character in the whole sequence $Y$. Finally, each $y$ is grouped as the output sequence $\hat{Y}$.

*3.2. Mixup in Speech-Transformer*

A robust end-to-end speech recognition model is insensitive to noise. Transformer has become the most popular baseline model for sequence-to-sequence problems due to its ability of modeling long-term dependency, but the complexity of Transformer-based model is extremely high. The dependence on data volume of Transformer is much higher than that of recurrent neural network. Thus, overfitting becomes a severe problem for Transformer-based methods in a limited-data setting. Data augmentation is an efficient tool to improve the robustness of neural networks, especially for low-resource speech recognition.

Mixup is a prevailing data augmentation method for supervised learning tasks. It trains a model on a linear combination of pairs of inputs and their targets to make the model more robust to adversarial samples. In this setting, the model can achieve more accurate rate under mixed noise. Mixup is computed as follows:

$$
\begin{aligned}
X_{mix} &= \lambda \cdot X_i + (1 - \lambda) \cdot X_j \\
Y_{mix} &= \lambda \cdot Y_i + (1 - \lambda) \cdot Y_j
\end{aligned}
\tag{2}
$$

where $X_i, X_j$ are the input vectors, and $Y_i, Y_j$ are the corresponding targets. $X_i$ and $X_j$ are randomly sampled from dataset $D = \langle X, Y \rangle$. $\lambda$ is sampled by a *Beta* distribution $B(\alpha, \alpha)$ with $\alpha \in (0, \infty)$. The generated pair $\langle X_{mix}, Y_{mix} \rangle$ is added into training dataset $D$.

For classification problems, Mixup can effectively improve the robustness of the model by smoothing loss landscapes. However, Mixup cannot be directly applied in speech recognition because the length of audio differs from each other which makes it difficult to calculate by Equation (2). Another reason is that the target sequence of each audio is discrete, and the linear combination of discrete data is meaningless. These two issues need to be addressed for most sequence-to-sequence problems.

In order to apply Mixup to speech recognition, Mixup is modified at the input level and the loss level, respectively. For input, two raw audios are mixed at the frame level. Before mixture, the shorter input will be padded to the same length as the longer one. Thus, the length of augmented sample $X_{mix}$ equals $max(len_{X_i}, len_{X_j})$.

A loss level mixture is adopted by mixing two loss function regarding the output. In general, the CTC loss function and the Cross-Entropy (CE) loss function are commonly used in end-to-end speech recognition. CE is adopted in Transformer-based model. For Speech-Transformer, the output of Transformer decoder is sent to a softmax classifier. The result of softmax layer becomes one of the input of CE loss. CE loss is calculated by Equation (3).

$$
\mathcal{L}_{CE}(\hat{Y}, Y) = -\frac{1}{N} \sum_{i=1}^{N} y_i \cdot \log \hat{y}
\tag{3}
$$

where $\hat{y}$ denotes the output of ASR model for each character, and $N$ is the length of $Y$.

The frame sequence and target are aligned by attention mechanism in Speech-Transformer which makes the output of softmax layer synchronized with label sequence.

After integrating Mixup into *CE*, the mixture of the *CE* loss becomes:

$$\mathcal{L}_{CE}(\hat{Y}, Y_{mix}) = \lambda \cdot \mathcal{L}_{CE}(\hat{Y}, Y_i) + (1 - \lambda) \cdot \mathcal{L}_{CE}(\hat{Y}, Y_j) \tag{4}$$

where $\lambda$ is the same weight as in the input.

**Theorem 1.** *For cross-entropy loss function, the mixture of the labels equals to the mixture of the CE loss.*

**Proof of Theorem 1.**

$$\begin{aligned}
\mathcal{L}_{CE}(\hat{Y}, Y_{mix}) &= \mathcal{L}_{CE}(\hat{Y}, \lambda Y_i + (1 - \lambda)Y_j) \\
&= -\sum (\lambda Y_i + (1 - \lambda)Y_j) log \hat{Y} \\
&= -\sum [\lambda Y_i log \hat{Y} + (1 - \lambda)Y_j log \hat{Y}] \\
&= \lambda \mathcal{L}_{CE}(\hat{Y}, Y_i) + (1 - \lambda)\mathcal{L}_{CE}(\hat{Y}, Y_j)
\end{aligned}$$

□

The mixed *CE* loss, respectively calculates the *CE* loss $\mathcal{L}_{CE}(\hat{Y}, Y_i)$ and $\mathcal{L}_{CE}(\hat{Y}, Y_j)$ with label sequence $Y_i$ and $Y_j$. Then, $\mathcal{L}_{CE}(\hat{Y}, Y_i)$ and $\mathcal{L}_{CE}(\hat{Y}, Y_j)$ are linearly combined with $\lambda$. This method is equivalent to interpolating the labels $Y_i$ and $Y_j$ directly.

*3.3. MKD: Mixup-Based Knowledge Distillation*

Knowledge distillation is commonly adopted in model compression for speech recognition. Knowledge distillation utilizes a teacher–student network structure that exploits soft labels from the teacher model to guide student network learning. However, this framework is subject to the amount of available data. In particular, tasks with fewer samples provide less opportunity for the student model to learn from the teacher. Even with a well-designed loss function, the student model is still prone to overfitting and effectively mimicking the teacher network on the available data. Existing data augmentation have been explored to combine with teacher–student network, improving the efficiency of knowledge distillation. Unlike other methods of data augmentation, Mixup is a data-agnostic approach which means no prior knowledge is required for augmentation. This brings an convenience for low-resource speech recognition to generate task-specific data.

Labels generated by Mixup are smoother than original one-hot labels. However, these soft labels don't include mutual information between each category. On the other hand, soft labels generated from teacher network could reflect the relationship between similar labels, which have more mutual information than one-hot encoding. The two arguments above inspired us to integrate Mixup into teacher–student framework to improve the data efficiency of knowledge distillation.

In our teacher–student framework, the architecture of student model $Model_S$ is the same as teacher model $Model_T$. There is only a difference in parameter scale between teacher model and student model in homogeneous neural networks. First, the input $X_i$ goes through the teacher model $Model_T$. The teacher model is trained by original CE loss.

The output of Softmax layer in $Model_T$ serves as the soft label of each frame. The student model is encouraged to imitate the prediction from teacher network by minimizing the *KL* distance:

$$\mathcal{L}_{KL} = \sum_{i=1}^{n} p(X_i) log \left( \frac{p(X_i)}{q(X_i)} \right) \tag{5}$$

Considering that the original cross-entropy is helpful for training the student network. The student network is trained with *CE* loss in usual way as well. The final loss $\mathcal{L}_{KD}$ of student model is the linear combinations of *CE* loss and *KL* loss:

$$\mathcal{L}_{KD} = \gamma \cdot \mathcal{L}_{KL} + (1 - \gamma) \cdot \mathcal{L}_{CE} \tag{6}$$

For softmax-level knowledge distillation, the distillation loss $\mathcal{L}_{KD}^k$ is calculated for *k*-th frame, and $\mathcal{L}_{KD}$ is composed of the accumulation of each frame loss.

When optimizing the teacher–student framework by Mixup, the label sequences $Y_i$, $Y_j$ are served as one of the input to the Decoder of the teacher model to obtain their soft labels $\hat{Y}_{t_i}, \hat{Y}_{t_j}$, respectively. As shown in Figure 1, when training the student network, the output of student network $\hat{Y}_{s_i}, \hat{Y}_{s_j}$ are used to calculate the $\mathcal{L}_{CE}$ and $\mathcal{L}_{KL}$. Equation (4) is applied to generate $\mathcal{L}_{CE}$ for each frame. However, the mixture of $\mathcal{L}_{KL}$ is different from that of $\mathcal{L}_{CE}$ because the KL-divergence $\mathcal{L}_{KL}$ is not linear for the label $Y$. To solve this problem, a novel loss function $\mathcal{L}_{MKL}$ is proposed to approximate $\mathcal{L}_{KL}$.

$$\mathcal{L}_{MKL} = \lambda \cdot \mathcal{L}_{KL}(\hat{Y}, Y_i) + (1 - \lambda) \cdot \mathcal{L}_{KL}(\hat{Y}, Y_j) \tag{7}$$

It could be proved that $\mathcal{L}_{MKL}$ is an upper bound on the $\mathcal{L}_{KL}$ using the properties of convex functions. The proof of Equation (7) is given below.

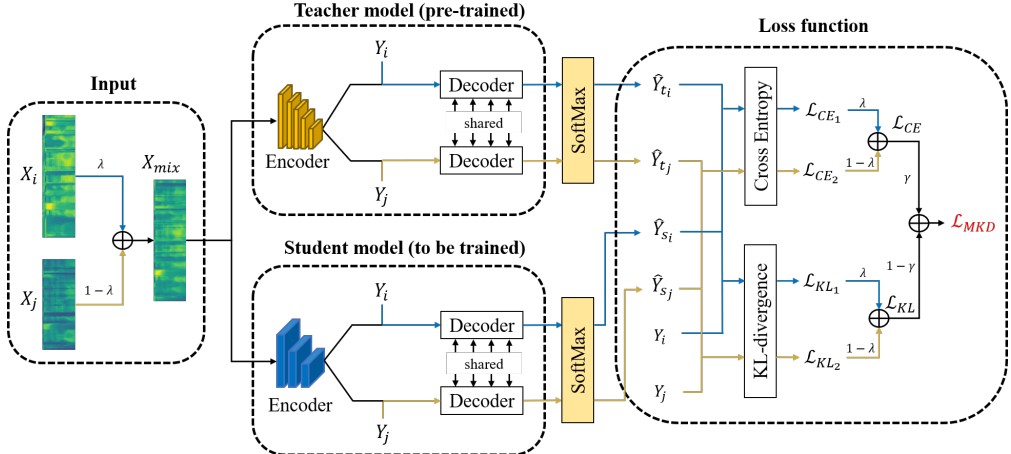

**Figure 1.** The flow chart of MKD. Two audios are mixed with $\lambda$ at the input stage. Then, $X_{mix}$ is fed to the teacher and student encoder, respectively. In the decoder module, the label sequences corresponding to each of the two audios are served as another part of the decoder input. Finally, the loss-level mixture is applied to correspond to the modification of the decoder.

**Theorem 2.** *The upper bound on the KL-divergence of $Y_{mix}$ is equivalent to the mixture of the KL-divergence.*

**Proof of Theorem 2.**

$$\mathcal{L}_{KL}(\hat{Y}, Y_{mix}) = \mathcal{L}_{KL}(\hat{Y}, \lambda Y_i + (1 - \lambda)Y_j)$$
$$= \mathcal{L}_{CE}(\hat{Y}, \lambda Y_i + (1 - \lambda)Y_j) + \sum [\lambda Y_i + [1 - \lambda]Y_j]log[\lambda Y_i + [1 - \lambda]Y_j]$$
$$\leq \lambda \mathcal{L}_{CE}(\hat{Y}, Y_i) + (1 - \lambda)\mathcal{L}_{CE}(\hat{Y}, Y_j) + \sum \lambda Y_i log \lambda Y_i + \sum (1 - \lambda)Y_j log(1 - \lambda)Y_j$$
$$= \lambda \mathcal{L}_{KL}(\hat{Y}, Y_i) + (1 - \lambda)\mathcal{L}_{KL}(\hat{Y}, Y_j) + \lambda log \lambda \sum Y_i + (1 - \lambda)log(1 - \lambda)\sum Y_j + C'$$
$$\Rightarrow \lambda \mathcal{L}_{KL}(\hat{Y}, Y_i) + (1 - \lambda)\mathcal{L}_{KL}(\hat{Y}, Y_j)$$

□

After obtaining $\mathcal{L}_{MKL}$ and $\mathcal{L}_{CE}$, a factor $\gamma$ is applied to balance two loss functions. The final loss $\mathcal{L}_{MKD}$ is the combination of *KL*-divergence and *CE* loss:

$$\mathcal{L}_{MKD} = \gamma \cdot \mathcal{L}_{MKL} + (1 - \gamma) \cdot \mathcal{L}_{CE} \tag{8}$$

## 4. Experimental Settings

This section introduces the experimental settings, including: the dataset, the evaluation indicators and the hyperparameters, respectively.

### 4.1. Dataset

Our experiments are conducted on a public Mandarin speech corpus named Aishell-1. The training set contains 120,098 speeches (about 150 h) recorded by 340 speakers. The development set contains 14,326 audios (approximately 40 h). In addition, 7176 voices (about 10 h) make up the test set. This corpus contains 4230 Chinese characters.

### 4.2. Performance Metrics

For the Mandarin dataset, we measured the character error rate (CER) and relative error rate reduction (RERR). This is because a single character often represents a word for the Mandarin writing system. To calculate CER, the number of errors is obtained by counting the substitutions, insertions, and deletions that occur in the recognition result. Then, it is divided by the total number of characters in the correct sentence. RERR shows how much the CER is reduced in proportion, compared to another method.

### 4.3. Model Settings

Experiments on Speech-Transformer has been performed. The 80 log-mel filter bank features are extracted by Kaldi toolkit [43]. Before training, low frame rate is applied for self-attention module to compute the similarity of each pair of frames. The mLFR processing, feature stacking and downsampling produce more sparse but more informative features. In our implementation, features are stacked with 4 frames to the left and skipped with 3 frames. The teacher model contains 6 Transformer encoder layers and 6 Transformer decoder layers. Each layer has 8 attention heads and a width of 512. The dimension of inner feed forward layer is 2048. During training, Adam [44] optimizer ($\beta_1 = 0.9$, $\beta_2 = 0.98$, $\alpha = 1 \times 10^{-9}$) is adopted. The epoch is 150. In each epoch, all the samples in dataset are shuffled to eliminate the effects of input order. Considering the varying length of the audio, a dynamic batchsize is applied in the experiment. Each batch consists of no more than 10,000 frames of audios in total length. A warm up strategy is employed at the first 4000 batches. For inference, the beam search with a beam size of 5 is performed.

For Mixup strategy, the mixed spectrogram is generated with $\lambda \sim B(0.5, 0.5)$. The proportion of mixed spectrogram in the whole dataset is $p$. Different strategies of training are explored in our experiments. Firstly, all the samples in dataset consists of mixed samples. Secondly, half of the samples are mixup samples. Another strategy is setting $p$ to 0.25. In practice, the factor $\sigma \sim U(0, 1)$ is set to control the proportion. If $\sigma > p$, the current batch is composed of mixed samples. Otherwise, the current batch consists of raw spectrogram.

The proposed MKD is a homogeneous knowledge distillation method. The framework of student model under MKD is the same as the teacher model. Two scale of student models are designed to testify the efficiency of MKD, 3 encoders with 3 decoders and 2 encoders with 2 decoders, respectively. The settings of student models are shown in Table 1. The student model calculates $\mathcal{L}_{KL}$ according to the output of softmax in teacher model. The weight of loss function $\gamma$ is 0.9. The training and inference of neural network are conducted on a RTX 3090 GPU.

**Table 1.** The settings of two student models.

| Properties | Stu1 | Stu2 |
|---|---|---|
| Encoder | 3 | 2 |
| Decoder | 3 | 2 |
| Head | 8 | 8 |
| Head size | 64 | 64 |
| Feed-forward | 2048 | 2048 |
| Parameter | 20 M | 10 M |
| Compression | 50% | 75% |

## 5. Results

Experiments on hyperparameter search are first conducted in this section. The experimental results of MKD are shown next. Finally the experiments of data efficiency are introduced to verify the validity of MKD.

### 5.1. Results of Hyperparameter Search

In order to improve the training efficiency, the proper hyperparameter of Mixup is searched at the first stage. No previous studies have experimented with Mixup on Mandarin datasets. Therefore, it is necessary to conduct experiments of hyperparameter tuning.

The Baseline is trained in usual way. When implementing Mixup strategy, several types of tricks are explored. Firstly, the value of $\alpha$ is searched from $0, 0.3, 0.5$. Second, label smoothing (LS) is tried as well. The label smoothing is conducted on the raw one-hot label. However, our experiments show that label smoothing hurts the Mixup strategy seriously not only in CER but also in the stability of model. In Table 2, the precision of model with label smoothing and $\alpha = 0.5$ decreases, even worse than the Baseline. Tried several more times, but the result is still the same. This phenomenon also appears in model with label smoothing and $\alpha = 0.3$. However, the impact is not serious. Thus, label smoothing is no longer used in the following experiments.

**Table 2.** Hyperparameter search of Mixup in Speech-Transformer on Aishell-1.

| Method | $\alpha$ | $p$ | CER |
|---|---|---|---|
| Baseline | 0.0 | 1.0 | 10.7% |
| +Mixup | 0.3 | 1.0 | 9.2% |
| +Mixup | 0.5 | 1.0 | 9.7% |
| +Mixup | 0.3 | 0.25 | 9.8% |
| +Mixup | 0.5 | 0.25 | **8.6%** |
| +Mixup | 0.3 | 0.5 | 9.2% |
| +Mixup | 0.5 | 0.5 | 8.8% |
| +Mixup + LS | 0.3 | 1.0 | 11.7% |
| +Mixup + LS | 0.5 | 1.0 | 10.1% |

The last two rows of Table 2 are two experiments of the proportion of mixed samples in the whole dataset. The result shows that both of two training strategies could achieve desirable performance. The model with $p = 0.5$ behaves better the model with $p = 0.25$, but the gap is very close. Compared with Baseline, both of them obtains 17% relative improvement.

In the following experiments, the hyperparameter of Mixup strategy is: $\alpha = 0.5$, no label smoothing and $p \in \{0.25, 0.5\}$.

### 5.2. Results of MKD

In order to testify the performance of proposed method, MKD is compared with previous softmax-level knowledge distillation (S-KD) and sequence-level knowledge distillation (SEQ-KD). The S-KD trains the model by Equation (6) directly and the training procedure

of SEQ-KD is the same as [45]. The Baseline is chosen as the teacher model. The size of Stu1 is 50% of teacher model, and the size of Stu2 is one third of teacher model. To make the results comparable, the teacher model and the student models are the same as the original one when training by S-KD, SEQ-KD and MKD. The Mixup strategy is adopted under previous settings.

The results of MKD are shown in Table 3. The proposed MKD beats the existing method by 1.7% and 0.4% compared to S-KD and SEQ-KD, respectively. MKD is more effective for small-scale models when the proportion of augmented samples occupies 25%. MKD can achieve a lowest character error rate of 9.2% on Stu1 with $p = 0.5$, which is even better than the result of the teacher model. Such phenomenon demonatrates that Mixup improves the generalization of ASR model. As the number of model parameters decreases, the effect of MKD also decreases. The experiments also indicates that data augmentation has a potential to be combined with knowledge distillation.

**Table 3.** The CER of proposed MKD on Aishell-1.

| Method | Stu1 | Stu2 | Tea |
|---|---|---|---|
| Baseline | 11.6% | 13.3% | 10.7% |
| +S-KD | 10.9% | 12.2% | - |
| +SEQ-KD | 11.4% | 12.8% | - |
| +MKD ($p = 0.25$) | 9.8% | **11.8%** | - |
| +MKD ($p = 0.5$) | **9.2%** | 12.8% | - |

Table 4 exhibits the RERR of proposed MKD compared with S-KD. For MKD with $p = 0.25$, the relative improvement is 11.0% in Stu1 and 3.3% in Stu2. For MKD with $p = 0.5$, the relative improvement reaches 15.6% in Stu1. However, the CER increases in Stu2 when $p = 0.5$, the reason is that the small parameter scale leads the underfitting problem. The number of mixed samples is required to be controlled for small models.

**Table 4.** The RERR of proposed MKD compared with S-KD.

| Method | p | Stu1 | Stu2 |
|---|---|---|---|
| MKD | 0.25 | 11.0% | 3.3% |
| MKD | 0.5 | 15.6% | −4.9% |

*5.3. Ablation Analysis*

In this section, ablation experiments are conducted for MKD. The parameter $\gamma$ plays a crucial role in balancing the proportion of soft and hard label contributions. Two sets of experiments are conducted at $p = 0.25$ and $p = 0.5$ for testifing the effect of $\gamma$, respectively. The range of $\gamma$ is $\{0.2, 0.5, 0.9\}$. Table 5 describes the results when $p = 0.25$.

**Table 5.** The effect of $\gamma$ in MKD when $p = 0.25$.

| $\gamma$ | Stu1 | Stu2 |
|---|---|---|
| 0.2 | 10.3% | 12.3% |
| 0.5 | **9.7%** | 12.2% |
| 0.9 | 9.8% | **11.8%** |

Experiments have shown that the distillation effect slowly diminishes as $\gamma$ decreases. On Stu1, the word error rate reaches 9.7% for $\gamma = 0.9$, while the rate drops by 0.5% for $\gamma = 0.9$. The results on Stu2 are similar as shown in the third column of Table 5. The $\gamma$ indicates the weight of the soft label. The larger the $\gamma$ is, the more information is retained. The results of the ablation experiments are consistent with the theoretical results.

The results of the ablation experiments at $p = 0.5$ are shown in Table 6. The results show that the character error rate changes from 9.2% to 10.4% when $\gamma$ is gradually reduced,

indicating that the distillation effect slowly diminishes. This finding is similar to the result when $p = 0.25$. In contrast to the results for $p = 0.5$, the MKD becomes more robust on the large model at $p = 0.25$, and the student model is less subject to the change of $\gamma$.

**Table 6.** The effect of $\gamma$ in MKD when $p = 0.5$.

| $\gamma$ | Stu1 | Stu2 |
|---|---|---|
| 0.2 | 10.4% | 12.1% |
| 0.5 | 10.2% | **11.8%** |
| 0.9 | **9.2%** | 12.8% |

*5.4. Data Efficiency Analysis*

In order to evaluate the effect of MKD on the data efficiency of the speech recognition model, experiments are conducted with different data volumes by dividing the training set into 10%, 50% and the full data set. In order to exclude the influence of other variables, other parameters are fixed during training, taking $p = 0.25$, $\lambda = 0.5$, $\gamma = 0.9$ for MKD. The size of the training set is represented by the variable $c$, and $c = 10\%$ indicates that 10% of the original training set is used as the current training set. The results of the experiments using the MKD are shown in Table 7.

**Table 7.** The CER of MKD in different dataset scales.

| Methods | c | Stu1 | Stu2 |
|---|---|---|---|
| Base | 10% | 36.8% | 45.7% |
| Base | 50% | 14.1% | 17.2% |
| Base | 100% | 11.6% | 13.3% |
| S-KD | 10% | 37.2% | 46.0% |
| S-KD | 50% | 13.8% | 15.9% |
| S-KD | 100% | 10.9% | 12.2% |
| MKD | 10% | **26.7%** | **36.9%** |
| MKD | 50% | **11.6%** | **14.9%** |
| MKD | 100% | **9.2%** | **11.8%** |

In the case of small dataset, the MKD method generally improves the recognition precision of ASR models compared to S-KD. When using 10% of the data, the distillation effect of the S-KD is rather inferior to that of the directly trained obtained models, with approximately 0.4% away from Baseline for both Stu1 and Stu2. However, the models trained with MKD can significantly improve the models, achieving 28.2% and 19.8% relative improvment. The gain of MKD is more significant when using 50% of the data, achieving relative gains of 15.9% and 6.3% on the two student models. The relative promotion on the full data set is 15.6% and 3.3%, respectively. The improvment of MKD becomes stronger as the amount of data decreased, indicating that MKD is a high data-efficient method that can extract informative feature with very small samples. The model trained on Stu1 using MKD on 50% of the data has the same CER as the Baseline trained using the full amount of data, demonstrating that the proposed MKD reduces the data dependence by at least half.

## 6. Discussion

We performed model compression experiments for the Mandarin ASR model. However, existing papers have either different model structures or different datasets. In addition, none of them performed the experiments of data efficiency. Therefore, it's difficult to compare with other research directly. To overcome this problem, the S-KD is reproduced with the same dataset. The results show that MKD achieves a lower CER against S-KD which demonstrates that MKD is a high-efficient knowledge distillation method.

The softmax-level and representation-level distillation methods are prevailing methods for attention-based ASR models. The representation-level distillation encourages the

student model to imitate the feature map of the teacher model. The combination with softmax-level distillation has been shown to be effective in the literature [46]. In our study, more attention are paid to the softmax-level knowledge distillation. The Mixup feature improves the efficiency of softmax-level distillation. However, whether the mixture of representation does harm to the model remains a mystery. How to integrate MKD into representation-level distillation is the problem to be fix in the future. Further, the possibility of joint representation-level and softmax-level distillation is another problem to be explored.

On the other hand, some people have researched the distillation between different network architectures such as from RNN to CNN. The conventional method in knowledge distillation has a limit that the student model structure should be similar to that of the given teacher model. Our experiments have verified the distillation of homogeneous model. The non-homogeneous model transfer is a promising research topic. Since CNN or RNN models have different model capacity from the Transformer, how to minimize the gap between them is a tough issue.

## 7. Conclusions

In order to improve the efficiency of knowledge distillation especially in the low-resource setting, MKD is proposed by integrating Mixup into softmax-level knowledge distillation framework. A loss-level mixture is adopted to address the discrete data of character label sequence in speech recognition. The mixed loss function $\mathcal{L}_{MKD}$ is presented to solve the problem caused by the non-linearity of label in the original KL-divergence when applying Mixup to knowledge distillation. It is theoretically shown that optimizing $\mathcal{L}_{MKD}$ is equivalent to optimize an upper bound of the original knowledge distillation loss. Experiments on Aishell-1 prove the effectiveness and efficiency of the proposed MKD. It obtains a 15.6% and 3.3% relative improvment on two student models with different parameter scales compared with the existing distillation method. Meanwhile, MKD decreases the demand of samples by one time in training Speech-Transformer.

Though MKD improves the performance of knowledge distillation of ASR model, there are some issues that need to be further tackled: (1) the applicability of MKD on other languages should be verified in the future. (2) The generalization of MKD is required to be explored further. The proposed MKD is not only a method in model compression, but a general framework in knowledge distillation. There are much more fields for MKD to play a role. The semi-supervised learning is the place which could make use of MKD most likely. Previous methods are still fragile under the few-shot setting. MKD may alleviate this problem for its expansion of the existing distribution.

**Author Contributions:** Conceptualization, X.W.; methodology, X.W.; software, Y.J.; formal analysis, Y.J.; investigation, Y.J.; data curation, Y.J.; writing—original draft preparation, Y.J.; writing—review and editing, X.W., J.W., Q.Q. In addition, Y.G.; visualization, Y.J.; supervision, X.W., J.W., Q.Q. In addition, Y.G.; project administration, X.W.; funding acquisition, X.W. All authors have read and agreed to the published version of the manuscript.

**Funding:** This work is supported by the National Natural Science Foundation of China (Grant No. 62172267), the National Key R&D Program of China (Grant No. 2019YFE0190500), the Natural Science Foundation of Shanghai, China (Grant No. 20ZR1420400), the State Key Program of National Natural Science Foundation of China (Grant No. 61936001), the Shanghai Pujiang Program (Grant No. 21PJ1404200), the Key Research Project of Zhejiang Laboratory (No. 2021PE0AC02).

**Institutional Review Board Statement:** Not applicable.

**Informed Consent Statement:** Not applicable.

**Data Availability Statement:** Data is contained within the article.

**Conflicts of Interest:** The authors declare no conflict of interest.

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
