# Peer review of "MKD: Mixup-Based Knowledge Distillation for Mandarin End-to-End Speech Recognition"

_algorithms, doi:10.3390/a15050160_

Round 1

Reviewer 1 Report

The approach taken by the authors is clear. The narration is a bit ongoing. Setting up a roadmap at the beginning of each section would help the reader. Also, most model settings in Section 4.3 are taken without justification or support in the literature.

I have a major concern regarding the data augmentation by interpolation Mixup. Since it concerns spectrograms, are we sure that the weighted sum of two spectrograms of words is still a valid data point for speech, i.e., can the resulting spectrogram be considered representative of a voice sample?

Since the analysis is conducted for Mandarin only, and Mandarin has its peculiarities, I guess that this should be reflected in the title.

The sections on datasets and performance metrics should be taken out of the Results section.

Some language slips. 

"The spectrogram is served" --> "The spectrogram serves"

" the complexity of ttransformer-based model are.." --> "the complexity of transformer-based model is..."

The authors use the term shape to refer to the spectrogram. Shape is used for a matrix just to mean whether it is square or rectangular; maybe the authors meant size.

Reviewer 2 Report

I recommend to make the abstract more concise, there is an unnecessarily broad introduction.
Keywords are chosen appropriately.
Line 29 The statement is true, but I would recommend basing it on the literature. 
The literature review is well conceived and the main ideas and opinions are reflected
I would recommend forming the usual Methodology and Aim section from some of the text in Chapters 3 and 4
The results are well described, the number of hours is problematic and could be better justified, for example against other research. 
The discussion section does not fully deal with the comparison of views and similar research, I would recommend additions. 
The results are well summarised, but I would also appreciate mention of the limitations of this research and an indication of future directions for further research in the conclusion. 
